# MmWave Radar and Vision Fusion for Object Detection in Autonomous Driving: A Review

**DOI:** 10.3390/s22072542

**Published:** 2022-03-25

**Authors:** Zhiqing Wei, Fengkai Zhang, Shuo Chang, Yangyang Liu, Huici Wu, Zhiyong Feng

**Affiliations:** 1Key Laboratory of Universal Wireless Communications, Ministry of Education, School of Information and Communication Engineering, Beijing University of Posts and Telecommunications, Beijing 100876, China; zhangfk@bupt.edu.cn (F.Z.); changshuo@bupt.edu.cn (S.C.); yangyangl@bupt.edu.cn (Y.L.); fengzy@bupt.edu.cn (Z.F.); 2National Engineering Lab for Mobile Network Technologies, Beijing University of Posts and Telecommunications, Beijing 100876, China; dailywu@bupt.edu.cn

**Keywords:** autonomous driving, radar and vision fusion, radar and camera fusion, object detection, data level fusion, decision level fusion, feature level fusion, lidar, survey, review

## Abstract

With autonomous driving developing in a booming stage, accurate object detection in complex scenarios attract wide attention to ensure the safety of autonomous driving. Millimeter wave (mmWave) radar and vision fusion is a mainstream solution for accurate obstacle detection. This article presents a detailed survey on mmWave radar and vision fusion based obstacle detection methods. First, we introduce the tasks, evaluation criteria, and datasets of object detection for autonomous driving. The process of mmWave radar and vision fusion is then divided into three parts: sensor deployment, sensor calibration, and sensor fusion, which are reviewed comprehensively. Specifically, we classify the fusion methods into data level, decision level, and feature level fusion methods. In addition, we introduce three-dimensional(3D) object detection, the fusion of lidar and vision in autonomous driving and multimodal information fusion, which are promising for the future. Finally, we summarize this article.

## 1. Introduction

Main causes of traffic accidents are the complicated road conditions and a variety of human errors. Autonomous driving technology can leave the judgment of road conditions and human manipulation to the vehicle itself, which can improve traffic efficiency and driving safety. In recent years, with the significant development of artificial intelligence (AI), Internet of Things (IoT), and mobile communication, etc., autonomous driving has developed rapidly. There is no doubt that in the foreseeable future, autonomous driving vehicles will enter people’s daily life and become a common AI product. The earliest research on autonomous driving was in the 1980s when Defense Advanced Research Projects Agency (DARPA) organized the autonomous land vehicle (ALV) project [1]. In 2009, Google announced the establishment of a research team to start research on autonomous driving technology [2]. In 2013, Baidu launched research on “baidu nomancar”. In the same year, Audi, Nissan, and other traditional automobile manufacturers began to layout autonomous driving cars. However, until now, autonomous driving has not fully entered people’s life. The complex traffic environment has too many uncertainties, which discourage people from entrusting their safety to autonomous driving vehicles.

In 2014, the Society of Automotive Engineers (SAE) released j3016 standard, which defined the standards of autonomous driving level from L0 (no driving automation) to L5 (full driving automation) [3]. Levels L4 and L5 can completely get rid of manual operation during driving. However, there are more challenges to guarantee the safety of driving [4]. For safety reasons, the anti-collision function is an essential part of automated driving system (ADS), and the core task of the anti-collision system is obstacle detection. Therefore, one of the challenges for high-level autonomous vehicles is accurate object detection in complex scenarios.

Object detection is the most popular research area in the field of vision image processing. In the early stage of research, due to the lack of efficient image feature representation methods, object detection algorithms relied on the construction of manual features. Representative traditional algorithms included histograms of oriented gradients (HOG) [5], deformable parts model (DPM) [6], and so on. In recent years, as deep learning has developed, its great potential in object detection has been discovered. Detection algorithms based on deep learning are divided into two types: one-stage and two-stage. The former contains YOLO [7], SSD [8], and RetinaNet [9], and the latter contains R-CNN [10], Fast R-CNN [11], and Faster R-CNN [12], etc. However, the current visual object detection algorithms have hit a performance ceiling because the detection algorithms face very complicated situations in practice. For the scenarios of autonomous driving, the obstacles include pedestrians, cars, trucks, bicycles, and motorcycles, and obstacles within the visual range have different scales and length-to-width ratios. In addition, there may be varying degrees of occlusion between obstacles, and the appearance of the obstacles may be blurred because of the extreme weather such as rainstorms, heavy snow, and fog, which result in a great reduction in detection performance [13]. Studies have shown that CNN has poor generalization ability for untrained distortion types [14].

Various studies have shown that a camera is not sufficient for autonomous driving tasks. Compared with vision sensors, the detection performance of millimeter wave (mmWave) radar is less affected by extreme weather [15,16]. In addition, the mmWave radar not only measures the distance, but also uses the Doppler effect of the signal reflected by the moving object to measure the velocity vector [17,18]. However, mmWave radar cannot provide the outline information of the object, and it is difficult to distinguish relatively stationary targets. In summary, the detection capabilities of the vision sensor and the mmWave radar could complement each other. The detection algorithms based on mmWave radar and vision fusion can significantly improve the perception of autonomous vehicles and help vehicles better cope with the challenge of accurate object detection in complex scenarios.

The process of mmWave radar and vision fusion based object detection is shown in Figure 1. The processes of mmWave radar and vision fusion consist of three parts: sensor selection, sensor calibration, and sensor fusion. In order to achieve the desired performance of object detection with mmWave radar and vision fusion, the following challenges need to be solved.

Spatiotemporal calibration: The premise of fusion is to be in the same time and space, which means that mmWave radar and vision information need to be calibrated.Information fusion: Object detection algorithms that fuse the sensing information of different sensors to achieve the optimal performance are essential.

In order to solve the above challenges, first, it is essential to analyze the characteristics of different sensors and select the appropriate sensor. Second, it is necessary to study the sensor calibration, which involves the coordinate transformation between different sensors, the filtering of invalid sensing information, and the error calibration. Last but not least, it is necessary to study the sensing information fusion and realize the improvement of sensing capability through the complement of different sensors, which involve the mmWave radar and vision fusion schemes.

In recent years, there have been some surveys on object detection, but most of them focus on visual solutions. Ref. [19] summarized the vision-based object detection solutions and conducted a comprehensive and detailed survey for object detection algorithms before and after the advent of deep learning. It is helpful for understanding vision-based object detection, but its detection targets were not only vehicles and there is no relevant content about fusion of radar and vision. Ref. [20] summarized the functions of each sensor module in the advanced driver assistance system (ADAS), but lacks the explanation of the fusion between radar and vision. Ref. [21] summarized the multi-sensor fusion and vehicle communication technology for autonomous driving, involving the fusion of cameras, mmWave radar, lidar, global positioning system (GPS), and other sensors and vehicle to everything (V2X) technology. It focused on the classification of sensor fusion schemes, but lacks sufficient and in-depth review on the radar and vision fusion. In this article, we focus on the low-cost mmWave radar and vision fusion solutions. The process of mmWave radar and vision fusion consists of sensor selection, sensor calibration, and sensor fusion, each of which is reviewed in detail. Specifically, we classified the fusion methods into three-level fusion schemes including data level, decision level, and feature level schemes.

As shown in Figure 2, the rest of this article is organized as follows. Section 2 introduces the tasks of object detection, evaluation criteria, and the public dataset. Section 3 reviews the sensor configuration schemes of some automobile manufacturers and the comparison of the sensors. Section 4 reviews the sensors calibration technologies. Section 5 reviews sensor fusion schemes. Section 6 analyzes the development trends of fusion of radar and vision for autonomous driving. Finally, we summarize this article in Section 7.

## 2. Tasks, Evaluation Criteria, and Datasets

### 2.1. Tasks

Object detection is of vital importance in the field of autonomous driving. In addition to target classification, the detection tasks include positioning of the existing objects in the input image.

The goal of two-dimensional (2D) object detection is to select the detected obstacle targets in the vision image of the vehicle with a 2D bounding box. The targets are then classified and positioned. The positioning here refers to the positioning of the targets in the image rather than the positioning of targets relative to the vehicle in the real world.

In three-dimensional (3D) object detection, the processes include the selection of the detected obstacle targets in the vision image of the vehicle with a 3D bounding box and the classification and positioning of the targets. The positioning is not only the positioning of the selected target in the image, but also determining the posture and position of the objects in the real world.

### 2.2. Evaluation Criteria

Average precision (AP) and average recall (AR), representing accuracy and regression in object detection, are commonly used as evaluation indices. The precision-recall (PR) curve can be obtained by taking recall values and precision values as the horizontal and vertical axes, respectively. The mean average precision (mAP), which represents the consolidated results of the detection model, can be obtained by calculating the average AP value of all classes.

Take the KITTI dataset, which is widely used in autonomous driving, as an example. For 2D object detection, the correctness of object positioning is determined by comparing whether the intersection over union (IoU) between the detection bounding box and the annotated bounding box is greater than the threshold [22]. It uses the mAP model proposed in [23] for performance evaluation. The calculation method is to set 11 equally spaced thresholds in [0, 1]. For recall value greater than each threshold, there is a maximum precision value, and the average value of 11 precise values is the final mAP value.

However, 3D object detection currently is more attractive in the studies of autonomous driving. The KITTI officially stipulated that for vehicles, correct prediction requires that the predicted 3D box overlaps the real 3D box by more than 70%, whereas for pedestrians and bicycles, 50% overlap of the 3D bounding box is required [24]. In addition, KITTI adopted the recommendations of the Mapillary team in [25], which proposed that 40 recall positions could calculate the mAP more accurately than the 11 recall positions method.

### 2.3. Datasets

Datasets are very important for the research of object detection. This section will briefly introduce some representative datasets for autonomous driving. Table 1 shows the basic features of some widely used datasets.

#### 2.3.1. Apolloscape

ApolloScape [26] is part of the Apollo Open Platform created by Baidu in 2017. It uses a Reigl lidar to collect point cloud. The 3D point cloud generated by Reigl is more accurate and denser than the point cloud generated by Velodyne. Currently, ApolloScape has opened 147,000 frames of pixel-level semantically annotated images, including perceptual classification and road network data, etc. These high-resolution images with pixel-by-pixel semantic segmentation and annotation come from a 10 km distance measurement across three cities. Moreover, each area was repeatedly scanned under different weather and lighting conditions.

#### 2.3.2. KITTI

The KITTI dataset [22], established by the Karlsruhe Institute of Technology in Germany and the Toyota American Institute of Technology in the USA, is currently the most commonly used autonomous driving dataset. The team used a Volkswagen car equipped with cameras and Velodyne lidar to drive around Karlsruhe, Germany for 6 hours to record the traffic information. The dataset provides the raw image and accurate 3D bounding box with class label for every sequence. The object classes include cars, vans, trucks, pedestrians, cyclists, and trams.

#### 2.3.3. Cityscapes

The Cityscapes dataset [27] is jointly provided by three German labs: Daimler, Max planck institute informatik, and Technische Universität Darmstadt. It is a semantic understanding image dataset of urban street scenes. It primarily contains 5000 high-quality pixel-level annotated images of driving scenes in an urban environment (2975 for training, 500 for validation, 1525 for test, for 19 categories in total) from over 50 cities. In addition, it has 20,000 rough-annotated images.

#### 2.3.4. Waymo Open Dataset

The Waymo dataset [28] is an open source project of Waymo, an autonomous driving company under Alphabet Inc. It consists of labeled data collected by Waymo self-driving cars under various conditions, including more than 10 million miles of autonomous driving mileage data covering 25 cities. The dataset includes lidar point clouds and vision images. Vehicles, pedestrians, cyclists, and signs have been meticulously marked. The team captured more than 12 million 3D annotations and 1.2 million 2D annotations.

#### 2.3.5. nuScenes

The nuScenes dataset [29], established by nuTonomy, is the largest existing autonomous driving dataset. It is the first dataset that is equipped with the full autonomous vehicle sensors. This dataset not only provides camera and lidar data, but also contains radar data, and it is currently the only dataset with radar data. Specifically, the 3D bounding box annotation provided by nuScenes not only contains 23 classes, but also has 8 attributes including pedestrian pose, vehicle state, etc.

## 3. Sensor Deployment

This section introduces the sensor deployment schemes for autonomous driving vehicles. Through the analysis of the sensor deployment for vehicles equipped with autonomous driving systems launched by several major automobile manufacturers, it can be found that mmWave radar, lidar, and cameras are the main sensors for vehicles to detect the obstacles. In the following sections, the sensor deployment schemes including sensor configuration, sensor comparison, and selection, are reviewed in detail.

### 3.1. Sensor Configuration

As shown in Table 2, the overwhelming majority of automobile manufacturers have adopted a sensor configuration scheme combining radar and cameras. In addition to Tesla, other manufacturers have used the fusion sensing technology combining lidar, mmWave radar, and cameras. It can be concluded that the sensing solution using the fusion of radar and vision is the current mainstream trend in the field of obstacle detection for autonomous driving vehicles. The reason is that the radar and the camera have complementary characteristics. The specific details will be explained in Section 3.2.

### 3.2. Sensor Comparison

Ref. [36] investigated the advantages and disadvantages of mmWave radar, lidar, and cameras in various applications. This article compares the characteristics of the three sensors, as shown in Table 3. According to Table 3, three sensors have complementary advantages.

#### 3.2.1. mmWave Radar and Lidar

As a common and necessary sensor on autonomous driving vehicles, mmWave radar has the characteristics of long range detection, low cost, and detectability of dynamic targets. Due to these advantages, the vehicle’s sensing ability and safety have been improved [37]. Compared with lidar, the advantages of mmWave radar are mainly reflected in the aspects of coping with severe weather and low deployment cost [36]. In addition, it has the following advantages.

The mmWave radar can detect obstacles within 250 m, which is of vital importance to the security of autonomous driving, whereas the detection range of lidar is within 150 m [41].The mmWave radar can measure the relative velocity of the target vehicle based on the Doppler effect with the resolution of 0.1 m/s, which is critical for vehicle decision-making in autonomous driving [41].

Compared with mmWave radar, lidar has the following advantages [38,39].

Lidar has relatively higher angle resolution and detection accuracy than mmWave radar. Additionally, the mmWave radar data is sparser.The measurements of lidar contain semantic information and satisfy the perception requirements of advanced autonomous driving, which mmWave radar lacks.The clutter cannot be completely filtered out from mmWave radar measurements, leading to errors in radar signal processing.

Lidar applies laser beams to complete real-time dynamic measurement, establishing a 3D environment model, which supports the prediction of the surrounding environment and the position and velocity of targets. Due to the characteristics of the laser, its propagation is less affected by clutter, and the ranging is non-coherent and large-scale. According to [40], the detection range of lidar can reach more than 50 m in the longitudinal direction. The detection circle radius in the periphery of the vehicle reaches more than 20 m. The rear end of the vehicle reaches a detection range of more than 20 m, with high resolution of angular and range. However, the field of view (FOV) of lidar is limited.

The price is one of the factors limiting the massive application of lidar. However, with the development of lidar technology, its cost is showing a downward trend. The more important factor is that lidar is highly affected by bad weather conditions. When encountering heavy fog, rain, and snow, the attenuation of the laser is greatly increased, and the propagation distance is greatly affected, thereby reducing its performance [36].

#### 3.2.2. Radar and Camera

Radar is the best sensor for detecting distance and radial speed. It has “all-weather” capability, especially considering that it can still work normally at night. However, radar cannot distinguish color and has poor capability for target classification [36]. Cameras have good color perception and classification capabilities, and the angle resolution capability is not weaker than that of lidar [36]. However, they are limited in estimating speed and distance [40]. In addition, image processing relies on the powerful computing power of the vehicular chip, whereas the information processing of mmWave radar is not required. Making full use of radar sensing information can greatly save computing resources [36].

Comparing the characteristics of radar and cameras, it can be found that there are many complementary features between them. Therefore, the application of radar and vision fusion perception technology in the field of obstacle detection can effectively improve the perception accuracy and enhance the object detection capability of autonomous vehicles. Either mmWave radar or lidar and vision fusion are helpful. The advantages of the two fusion schemes come from the respective advantages of mmWave radar and lidar. In future research, the fusion of mmWave radar, lidar, and vision may have greater potential.

## 4. Sensor Calibration

Due to the difference in the spatial location and sampling frequency of different sensors, the sensing information of different sensors for the same target may not match. Hence, calibrating the sensing information of different sensors is necessary. The detection information returned by mmWave radar is radar points, and cameras receive visual images. We selected the camera and mmWave radar data from nuScenes [29] as an example. The data provided by this dataset have been processed by frame synchronization, so time synchronization is not required, and Figure 3 can be obtained through spatial coordinate transformation. The RGB value of the radar point is converted from the three physical quantities of transverse velocity, longitudinal velocity, and distance, and the color of the radar point represents the physical state of the object corresponding to the radar point. Generally speaking, sensor calibration involves coordinate calibration [42,43,44,45,46,47,48], radar point filtering [43,45], and error calibration [49,50,51].

### 4.1. Coordinate Calibration

The purpose of coordinate calibration is to match the radar points to the objects in the images. For coordinate calibration, the most common methods are classified into coordinate transformation method [45,46], sensor verification method [42,44,47], and vision based method [43,52], which are reviewed as follows.

Coordinate transformation method: The coordinate transformation method unifies the radar information and vision information under the same coordinate system through matrix operations. In [46], space calibration was completed by the method of coordinate transformation according to the spatial position coordinates of mmWave radar and vision sensors. For the time inconsistency caused by different sensor sampling rates, the thread synchronization method is adopted to realize the acquisition of the image frame and mmWave radar data simultaneously. Ref. [45] used the point alignment method based on pseudo-inverse, which obtains the coordinate transformation matrix by using the least square method. The traditional coordinate transformation cannot generate the accurate position of the target, which brings errors to the final results. In [53], Wang et al. proposed a calibration experiment to project the real coordinates into the radar detection map without special tools and radar reflection intensity, which weakens the dependence on calibration errors.Sensor verification method: The sensor verification method calibrates multiple sensors to each other with the detection information of different sensors on the same object. In [42], the sensor verification consists of two steps. First, the target list is generated by radar, and then the list is verified by the vision information. In [47], after the coordinate transformation of radar, the image is first searched roughly and then compared with the radar information. The result of the comparison divides the targets into two types: matched target and unmatched target. In [44], Streubel et al. designed a fusion time slot to match the objects detected by radar and vision in the same time slot.Vision based method: In [52], the motion stereo technology was used to achieve the matching of radar objects and image objects. In [43], Huang et al. used adaptive background subtraction to detect moving targets in the image, generate candidate areas, and verify the targets by judging whether the radar points are located in the candidate areas.

### 4.2. Radar Point Filtering

The purpose of radar point filtering is to filter out noise and useless detection results to avoid misjudgments caused by these radar points. In [45], Guo et al. proposed a method for noise filtering and effective target extraction using intra-frame clustering and inter-frame tracking information. In [43], the radar points were filtered by the speed and angular velocity information obtained by mmWave radar. The invalid radar points were then filtered, which reduces the impact of stationary targets such as trees and bridges on mmWave radar.

At present, in the field of autonomous driving, researchers mostly use open source datasets for training, and the objects they want to detect are generally vehicles or pedestrians. Therefore, the method of [43] can be applied to filter the radar points according to the speed and other information to exclude non-vehicle and non-pedestrian objects.

### 4.3. Error Calibration

Due to errors in sensors or mathematical calculations, there may be errors in the calibrated radar points. Some articles have proposed methods to correct these errors. In [50], a method based on interactive fine-tuning was proposed to make ultimate rectification to the radar points projected on the vision image. The authors in [51] proposed an improved extended Kalman filter (EKF) algorithm to model the measurement errors of different sensors. In [49], the influence of various coordinates on detection results was analyzed and discussed, and a semi-integral Cartesian coordinate representation method was proposed to convert all information into a coordinate system that moves with the host vehicle.

With the current use of open source datasets, error calibration is not required. However, if the dataset is self-made, radar filtering and error correction are necessary technical steps.

## 5. Vehicle Detection Based on Sensor Fusion

Generally speaking, there are three fusion levels for mmWave radar and vision, including data level, decision level, and feature level. Data level fusion is the fusion of data detected by mmWave radar and cameras, which has the minimum data loss and the highest reliability. Decision level fusion is the fusion of the detection results of mmWave radar and cameras. Feature level fusion requires extracting radar feature information and then fusing it with image features. The comparison of the three fusion levels is provided in Table 4.

### 5.1. Data Level Fusion

Data level fusion was a mature fusion scheme and has not been the mainstream research trend at present. However, its idea of fusing different sensor information is still useful for reference. As shown in Table 5, data level fusion first generates the region of interest (ROI) based on radar points [42,45,54,55]. The corresponding region of the vision image is then extracted according to the ROI. Finally, feature extractor and classifier are used to perform object detection on these images [45,47,53,55,56,57,58,59,60,61]. Some literatures use neural networks for object detection and classification [61,62]. For data level fusion, the number of effective radar points has a direct impact on the final detection results. If there is no radar point in a certain part of the image, this part will be ignored. This scheme narrows the searching space for object detection, saves computational resources, and leaves a hidden security danger at the same time. The data level fusion process is shown in Figure 4.

#### 5.1.1. ROI Generation

ROI is a selected area in the image, which is the focus of target detection. Compared with a pure image processing scheme, the data-level fusion scheme uses radar points to generate ROI, which can significantly improve the speed of ROI generation [42]. The size of the initial ROI is determined by the distance between the obstacle and mmWave radar [45]. The improved front-facing vehicle detection system proposed by [54] can detect overtaking with a high detection rate. This method focuses on the area where overtaking is about to occur. The overtaking is determined by checking two specific characteristics of vehicle speed and movement angle. In [55], a square with 3 meters of side length centered on the radar point is taken as the ROI.

#### 5.1.2. Object Detection

Because of the uncertainty of the position and size of the objects in the image, the object detection based on vision often adopts sliding window and multi-scale strategy, which produces a fair amount of candidate boxes, resulting in low detection efficiency. The mmWave radar and vision fusion scheme can avoid the sliding window method, which reduces the computational cost and improves the detection efficiency. Object detection tasks focus on the vision image processing, and the task can be divided into three steps: image preprocessing [45,53,56,57,61], image feature extraction [48,55,56,57,58,59,61,63,64], and object classification [47,56,60,61].

Image Preprocessing

In order to remove the noise in the image and enhance the feature information, so as to facilitate the subsequent feature extraction and object classification tasks, image preprocessing is necessary. The methods of image preprocessing mainly include histogram equalization, gray variance, contrast normalization, and image segmentation.

In [56], Bombini et al. conducted a series of tests on histogram equalization, grayscale variance, and contrast normalization in order to obtain the invariance under different illumination conditions and cameras. They concluded that contrast normalization achieved better performance. In [45], the gradient histogram method was used to preprocess the image, and an improved position estimation algorithm based on ROI was proposed, which can obtain a smaller potential object region and further improve the detection efficiency. In [61], median filtering, histogram equalization, wavelet transform, and Canny operator were used in image preprocessing.

In [53,57,61], radar points were taken as the reference center to segment the image, and then the object boundary was determined to improve the object detection speed.

Feature Extraction

The purpose of feature extraction is to transform the original image features into a group of features with obvious physical or statistical significance, which is convenient for object detection. In the stage of image feature extraction, the available vehicle features include symmetry and underbody shadow, etc.

In [58,59,63], symmetry was used for ROI detection and [48] utilized shadow detection to obtain feature information, whereas in [55,65], vertical symmetry and underbody shadow characteristics were comprehensively utilized for vehicle detection. In [55], the gradient information of the image was used to locate the boundary effectively, and the method based on gradient vector flow (GVF) Snake was used to describe the accurate contour of the vehicle. As the color distribution of the object was stable during the movement, the histogram matching method was feasible for tracking the vehicle.

In [64], Kadow et al. applied Haar-like model for feature extraction, which is a classic feature extraction algorithm for face detection. In order to improve detection rate, mutual information was used for Haar-like feature selection [56].

In [57], the visual selective attention mechanism and prior information on visual consciousness during human driving was proposed to extract features and identify object contours from segmented images. The shadow was extracted by histogram and binary image segment, and the pedestrian edge was detected by 3 × 3 image corrosion template.

Object Classification

In the object classification stage, Adaboost, support vector machine (SVM), and other object classification algorithms are used to select the final box of the vehicle in the vision image. In [56], the Adaboost algorithm is used to scan the ROI projected on the image plane. In [47,60], SVM was used for object recognition and classification. Ref. [66] combined ROI image and Doppler spectrum information for object classification. Ref. [61] adopted object classification based on infrared image analysis. The authors classified the objects into point objects and regional objects according to the object area, and used a neural network-based classifier to classify the regional targets [61]. Ref. [62] utilized multilayer in-place learning network (MILN) as a classifier, which demonstrated superior accuracy in two-class classification tasks.

### 5.2. Decision Level Fusion

Decision level fusion is the mainstream fusion scheme at present.The process is shown in the Table 6. The advantage of radar is longitudinal distance, and the advantage of the vision sensor is horizontal field of view. Decision level fusion can take into account the advantages of both aspects and make full use of sensing information. The challenge of the decision-level fusion filtering algorithm is to model the joint probability density function of the two kinds of detection information. This is due to the fact that the noise of the two kinds of detection information is different. The decision level fusion mainly includes two steps: sensing information processing [67,68,69,70,71,72] and decision fusion [65,68,73,74,75,76,77,78,79]. The decision level fusion process is shown in Figure 5.

#### 5.2.1. Sensing Information Processing

The processing of sensing information includes radar information and vision information. Radar detection results generate a list of objects and contain information such as the speed and distance of objects [67,68]. Vision information processing performs object detection algorithm on images, which includes traditional feature extraction combined with classifier [68,69] and convolutional neural network (CNN) [70,71,72].

Radar Information

A mmWave radar and vision fusion system was proposed for autonomous driving navigation and lane change in [67], in which the radar sensor obtains the target distance through fast Fourier transform (FFT), obtains the angular position of the target through digital wavefront reconstruction and beamforming, and then analyzes the target position. In [68], the results of mmWave radar detection is a list of possible objects in the radar FOV, and each element in the list includes the distance, azimuth angle, and relative velocity of the detected object.

Image Object Detection

In the visual detection of [68], a histogram was used to calculate edge information. Gradient histogram feature extraction and Boosting based classifier were then used to detect a pedestrian. Ref. [69] proposed an auxiliary navigation method combining mmWave radar and RGB depth sensor and used the MeanShift algorithm to detect objects in depth images. Furthermore, the average depth of ROI determined the distance of the detected object.

Ref. [70] has improved the vision data processing algorithm in [69], and applied mask R-CNN for object detection. In [71], YOLO V2 is used for vehicle detection, and the input is an RGB image of the size 224 × 224. Preprocessing subtracts the average RGB value of training set images from each pixel. Ref. [72] also applied YOLO V2 for vehicle detection and proved that YOLO V2 is superior to faster R-CNN and SSD in both speed and accuracy, which is more suitable for vehicle detection tasks. Ref. [80] applied YOLO V3 in obstacle detection, and its weight is pre-trained in the COCO dataset.

#### 5.2.2. Decision Fusion

The decision level fusion of vehicle detection fuses the detection results of different sensors. The mainstream filtering algorithms apply Bayesian theory [73,74], Kalman filtering framework [75,76,77], and Dempster Shafer theory [68]. In some literatures, the list of radar detection targets was used to verify vision detection results [78,79]. In addition, ref. [65] proposed the motion stereo algorithm to adjust and refine the final detection results.

Fusion Methods Based on Bayesian Theory

Ref. [73] proposed a method based on Bayesian theory to solve the problem of multi-sensor data fusion by using a probabilistic reasoning method, which is called Bayesian programming. The traditional multi-sensor fusion algorithms are no longer applicable when new sensors are added. The fusion algorithm is modularized and generalized in [74], and a dynamic fusion scheme based on a Bayesian network was proposed to improve the reusability of each fusion algorithm.

Fusion Methods Based on Kalman Filter

Based on the EKF framework of the Lie group, a decision level fusion filter using a special Euclidean group is proposed in [75]. Ref. [76] proposed a fusion framework that can track the detection object simultaneously in 3D space and a 2D image plane. An uncertainty driven mechanism similar to a Kalman filter is used to equalize the sensing results of different qualities. In [77], the given image was first detected by radar to roughly search the target. A trained spot detector was then used to obtain the object’s bounding box. The information fusion method based on Kalman filter was adopted, and the functional equivalence of centralized and decentralized information fusion schemes was proved.

Fusion Methods Based on Dempster Shafer Theory

Ref. [68] proposed decision level fusion based on the Dempster Shafer theory, taking the detection lists of multiple sensors as input, using one of them as a temporary evidence grid and fusing it with the current evidence grid, and finally performing clustering processing. The target was identified in the evidence grid.

Fusion Methods Based on Radar Validation

Ref. [78] overlapped the target list generated by vision detection and radar detection to generate a unique vehicle list. The radar data was used to verify the vision detection results. If there was a target matching the vision detection results in the radar data, a blue box would be marked as a strong hypothesis. Otherwise, if there was no target, it would not be discarded: a green box would be marked as a weak hypothesis. Ref. [79] proposed a multi-target tracking (MTT) algorithm that can correct the tracked target list in real time by evaluating the tracking score of the radar scattering center. The stereo vision information is used to fit the contour of the target vehicle, and the radar target matched with the target vehicle is used to correct its position.

### 5.3. Feature Level Fusion

Feature level fusion is a new scheme that has emerged in recent years.The process is shown in the Table 7. It is a common approach to use an additional radar input branch in feature level fusion methods [81,82,83,84,85]. The CNN-based objects detection model can effectively learn image feature information. By transforming radar detection information into image form, the detection model can learn radar and vision feature information simultaneously, and the feature level fusion will be realized. The feature level fusion process is shown in Figure 6.

#### 5.3.1. Object Detection Framework

Convolutional neural networks (CNNs) are widely applied in object detection based on feature level fusion. At present, some algorithms have achieved the results of superior performance, such as Faster RCNN, YOLO (V3), SSD, RetinaNet, etc.

Detection Framework Based on CNN

Faster-RCNN [12] is a widely used detector, and its integration has been greatly improved compared with Fast-RCNN [11]. Its main contribution is the introduction of the regional advice network (RPN), which enables nearly costless area suggestion. The RPN can simultaneously predict the target bounding box and categorization score for each location.

YOLO [7] is the first one-stage detector, which is the abbreviation of “You Only Look Once”. It adopts a new detection idea: using a single neural network to complete the detection task. It divides the vision image into multiple regions and predicts bounding box for each region. Therefore, the detection speed has been remarkably improved. However, its positioning accuracy is reduced compared with the two-stage detector.

SSD [8] is another excellent one-stage detector. It eliminates the bounding box suggestion and pixel or feature resampling stage and fundamentally improves the detection speed. The detections in different scales are generated from the feature maps of different scales, and the detections are clearly separated by aspect ratio, such that the accuracy is significantly improved. As a one-stage detector with high detection speed, SSD also ensures the detection accuracy close to that of two-stage detectors.

As to RetinaNet [9], the inventors discussed the reason why the accuracy of one-stage detector is lower than that of two-stage detector, and concluded that this phenomenon is caused by the extreme foreground background class imbalance. They proposed a new loss function named Focal loss to reshape the standard cross entropy loss, aiming to focus more attention on the objects that are difficult to classify during training.

Fusion Framework Based on CNN

In [81], feature level fusion was first applied in mmWave radar and vision fusion. Its detection network was improved on the basis of SSD [8]. The radar branch was concatenated after the second ResNet18 block of the vision branch.

In [82], a new sensor fusion framework called RVNet was proposed, which was similar to YOLO [7]. The input branches contain separate radar and vision branches and the output branches contain independent branches for small and large obstacles, respectively.

The CNN used in [83] was built on RetinaNet [9] with a VGG backbone [86], named CameraRadarFusionNet (CRF-Net). The author expanded the network so that it can process an additional radar input branch. Its input branches were radar feature and vision feature, and the output results are the 2D regression of the coordinates of the objects and the classification fraction of the objects.

In [84], the author proposed a new detection network called spatial attention fusion-based fully convolutional one-stage network (SAF-FCOS), which was built on FCOS [87]. The radar branch was improved from ResNet-50, and the vision image branch adopted a two-stage operation block similar to ResNet-50. In order to improve the detection accuracy, the final object detection comprehensively utilized FCOS [87] and RetinaNet [9].

#### 5.3.2. Radar Feature Extraction

The purpose of radar feature extraction is to transform radar information into image-like matrix information, because radar information cannot be fused with image information directly. Radar feature extraction mostly adopts the method of converting radar points to the image plane to generate a radar image. The transformed radar image with multiple channels contains all the environmental features detected by the radar. Each channel represents a physical quantity such as distance, longitudinal speed, lateral speed, and so on.

Ref. [85] proposed a new Conditional Multi-Generator Generative Adversarial Network (CMGGAN), which takes the measured data of radar sensors to generate artificial, camera-like environmental images, including all environmental features detected by radar sensors. A new description method for radar features was proposed in [82], which is called radar sparse image. Radar sparse image is a 416 × 416 three-channel image, whose size directly corresponds to the size of the vision image. The three channels contain radar point velocity and depth feature information. In [84], Chang et al. converted the depth, horizontal, and vertical information at the radar point into real pixel values of different channels. For the area without radar points, they set the pixel value to 0 and rendered the radar image with circles centered on radar points. Based on experimental results, the anti-noise performance of the radar image saved in the PNG format is better than that of the jpg format. In [83], considering the lack of height information in radar detection results, Nobis et al. stretched the projected radar points in the vertical direction in order to better integrate them with the image. The features of radar information were stored in the enhanced image in the form of pixel values. In addition, a ground truth noise filter was proposed to filter the invalid radar points.

#### 5.3.3. Feature Fusion

The basic feature fusion methods can be classified into two kinds: concatenation and element-wise addition. The former concatenates the radar feature matrix and the image feature matrix into a multi-channel matrix, whereas the latter adds two matrices into one.

In [81], two fusion methods of concatenation and element-wise addition are set, and the experimental results show that both fusion methods have improved the detection performance. The element-wise addition method performs better on manually labeled test sets, and the concatenation method performs better on generated test sets. Refs. [82,83] both adopt the concatenation method. In [84], a new block named spatial attention fusion (SAF) was proposed for sensor feature fusion. An SAF block was used to generate an attention weight matrix to fuse radar and vision features. At the same time, [84] compared the SAF method with three methods of element-wise addition, multiplication, and concatenation, and the result shows that SAF has the best performance. In addition, [84] conducted generalization experiments on faster R-CNN, and the detection performance was also improved by the SAF model.

## 6. Challenges and Future Trends

### 6.1. Challenges

For the object detection task, the current research results have achieved superior performance, however, most of these results are 2D object detection. In real autonomous driving scenarios, complex traffic environments often require 3D target detection to more accurately perceive environmental information. The performance of the current 3D object detection network is far below the level of 2D detection. Therefore, improving the accuracy of 3D object detection is not only a challenge in the field of autonomous driving, but also a major challenge in the task of object detection.

Challenges still exist for the fusion of mmWave radar and vision, which is the focus of this article. The biggest drawback of mmWave radar is the sparseness of radar features. Compared with vision images, mmWave radar provides very little information and cannot bring significant performance improvement. In addition, whether the feature information of mmWave radar and vision can be further integrated and the associated mutual information between them has been mined remains to be studied. Therefore, mmWave radar vision fusion still has two major challenges: sparse perception information and more efficient fusion methods, which are also the two major challenges in the field of multi-sensor fusion.

### 6.2. Future Trends

For the future development of object detection in the field of autonomous driving, the authors of this article believe that there are three main trends. One of them is 3D object detection. Improving the accuracy of 3D object detection will be a major research trend. The remaining two trends concern radar vision fusion. On the one hand, it is necessary to integrate new sensing information, namely, adding new sensors, such as lidar, which has achieved superior performance in autonomous driving; on the other hand, it is necessary to explore new ways of sensing information fusion, such as multimodal fusion.

#### 6.2.1. 3D Object Detection in Autonomous Driving

Three-dimensional object detection can not only identify the position of the object in the image, but can also detect the pose and other information of the object in the 3D space, which is more in line with the requirements of autonomous driving. In addition, the current mainstream application scenario of 3D object detection is autonomous driving, so that the research on object detection in the field of autonomous driving in the future will focus on 3D object detection. A common method for vision-based 3D object detection is to estimate the key points of the 3D bounding box within a 2D bounding box. Such works include SSD-6D [88], which is an SSD-based six-dimensional pose estimation network, and Mono3D [25], which uses semantic and instance segmentation information to score candidate boxes. Ref. [25] also chose two-stage 3D object recognition, and its main contribution is to propose a disentangling transformation to separate multiple parameters that need to be trained to improve the efficiency of training. Ref. [89] used a 3D anchor constraint box to directly obtain a 3D constraint box by regression and the gap between the anchor constraint box, in which a network using different convolution weights at different positions of the feature map was used to improve the performance. There are also methods that do not require 2D object detection first. These methods transform image features into new representations. Ref. [90] proposed orthographic feature transform to convert the general image feature map into a 3D voxel map, then performed normalization in the vertical dimension, converted it to a bird’s-eye feature map and regressed the 3D parameters on the bird’s-eye view.

3D object detection based on multi-sensor fusion also adds radar input branches and information fusion module on the basis of vision-based object detection network. Ref. [91] used a scheme similar to feature-level fusion, first rendering the radar points into a rectangular area through 2D detection and then performing 3D detection. In addition, since lidar is rich in features and can reconstruct object contours, it is easier to estimate 3D bounding boxes, there are more studies on multi-sensor fusion 3D object detection with lidar.

#### 6.2.2. Lidar in Autonomous Driving

As the cost of lidar decreases, autonomous driving vehicles equipped with lidar have become a trend. However, lidar is not a substitute for mmWave radar. As noted in Section 3, mmWave radar has its own unique advantages. Lidar has higher detection accuracy, and they complement each other’s advantages. The fusion of lidar and vision is becoming promising in autonomous driving. The main research includes object detection, object classification, and road segmentation.

Object Detection

The proposed fusion method in [92] first applied a lidar point cloud to generate suggestions for potential car positions in the image. The position of the bounding box was then refined by mining multiple layers of information in the proposal network. Finally, the object detection was performed by a detection network that shared part of the layers with the proposed network. Ref. [93] proposed a fusion method similar to the data level fusion of mmWave radar and vision. It utilized lidar instead of mmWave radar to generate initial recommendations for the position of objects. Multi-scale features were then exploited to realize accurate detection of objects of different sizes.

Object Classification

Ref. [94] presented a lidar and vision fusion method for object classification that combines Deep-CNN (DCNN) and upsampling theory. It can inherit the advantages of the two methods and avoid the shortcomings of each method. This method up-sampled the lidar information and converted it into depth information, which would be fused with the vision image data and sent into the DCNN for training. The experimental results showed that this method exhibits superior effectiveness and accuracy of object classification.

Road Detection

Ref. [95] proposed a method to combine lidar point clouds and vision images for road detection. First, the method of point alignment was used to project the lidar point cloud onto the vision image. The vision image was then upsampled to obtain a set of dense images containing spatially encoded information. Finally, they trained multiple fully convolutional neural networks (FCN) for road detection. The proposed FCN was evaluated on KITTI and got a MaxF score of 96.03%, which is one of the top-performing methods so far.

#### 6.2.3. Multimodal Information Fusion

The objects we see, the sounds we hear, and the smells we smell in the real world are all different modal information. Combining the multimodal information of the same thing can help AI better understand the thing. At present, the more popular research in this field involves the association of pictures, voices, videos, and texts, such as search engines.

Therefore, whether it is mmWave radar or lidar, its sensing information is the same environmental information in different modes. Radar sensing information and vision information are also information of different modalities. Considering radar vision fusion as multimodal information fusion, there may be a better solution.

In addition, in the field of autonomous driving, the mmWave radar data provided by the dataset are post-processed data. However, from the perspective of information conservation, the amount of information contained in the post-processed radar data must be lost relative to the original data. If the original radar detection data and vision images are regarded as two different modalities of sensing information to be fused, more abundant sensing information can be obtained.

The challenge of multimodal information fusion is how to perfectly combine the information of different modalities and the noise they carry and how to mine the relation information to assist the understanding of the same thing.

## 7. Conclusions

Object detection is one of the most important tasks for autonomous driving. In this article, we provide an overview of mmWave radar and vision fusion for vehicle detection. First, we introduced the tasks, evaluation criteria, and datasets of autonomous driving. Second, we divided the mmWave radar and vision fusion process into three parts and the fusion algorithms were classified into three levels: data level, decision level, and feature level. Finally, 3D object detection, lidar vision fusion and multimodal information fusion, as promising technology in the future, is reviewed. 

## Figures and Tables

**Figure 1 sensors-22-02542-f001:**
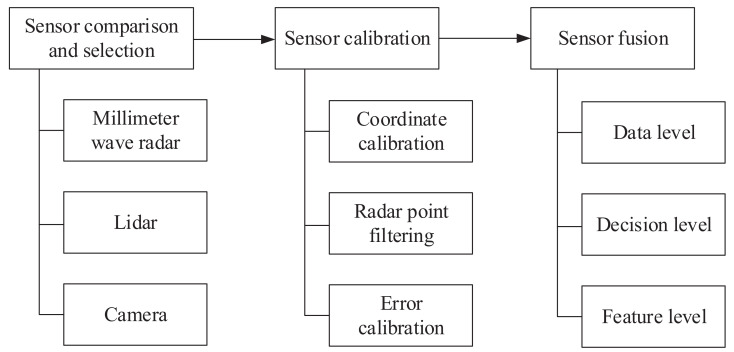
Process of mmWave radar and vision fusion.

**Figure 2 sensors-22-02542-f002:**
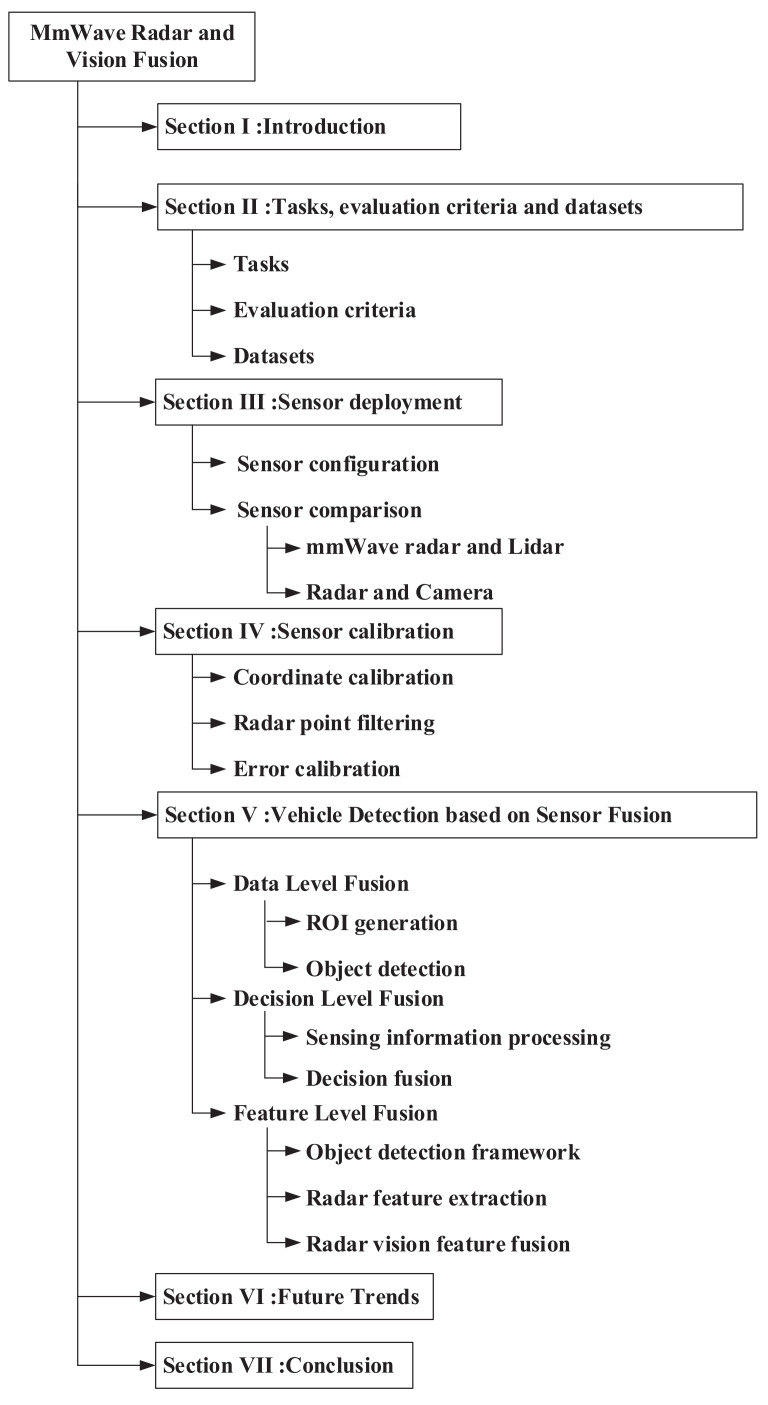
The organization of this article.

**Figure 3 sensors-22-02542-f003:**
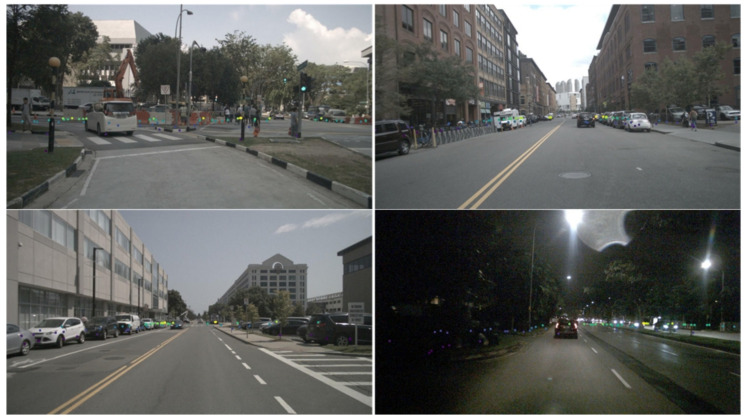
Radar points are projected on the image and rendered into different colors.

**Figure 4 sensors-22-02542-f004:**
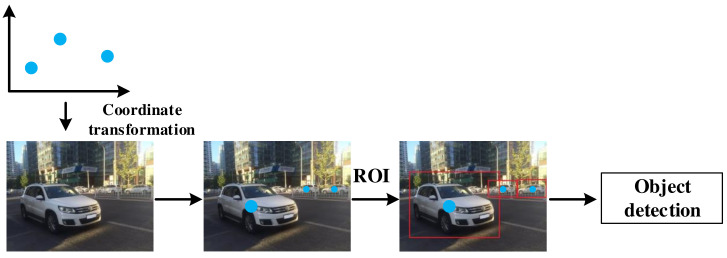
Data level fusion.

**Figure 5 sensors-22-02542-f005:**
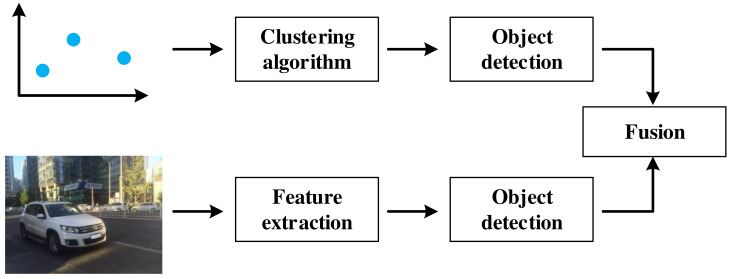
Decision level fusion.

**Figure 6 sensors-22-02542-f006:**
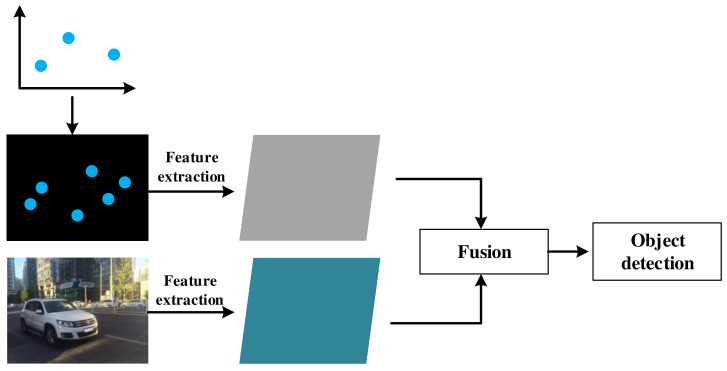
Feature level fusion.

**Table 1 sensors-22-02542-t001:** Autonomous driving datasets. “Y” indicates the existence of the sensing information in this dataset and “N” indicates the absence of the sensing information in this dataset.

Dataset	Release Year	RGB Image	Radar	Lidar
Apolloscape	2018	Y	N	Y
KITTI	2012	Y	N	Y
Cityscapes	2016	Y	N	N
Waymo Open Dataset	2019	Y	N	Y
nuScense	2019	Y	Y	Y

**Table 2 sensors-22-02542-t002:** Autonomous driving sensor solutions of some manufacturers [30,31,32,33,34,35].

Company	Autonomous Driving System	Sensor Configuration
Tesla	Autopilot	8 cameras, 12 ultrasonic radars, mmWave radar
Baidu	Apollo	Lidar, mmWave radar, Camera
NIO	Aquila	Lidar, 11 cameras, 5 mmWave radars, 12 ultrasonic radars
Xpeng	XPILOT	6 cameras, 2 mmWave radars, 12 ultrasonic radars
Audi	Traffic Jam Pilot	6 cameras, 5 mmWave radars, 12 ultrasonic radars, Lidar
Mercedes Benz	Drive Pilot	4 panoramic cameras, Lidar, mmWave radar

**Table 3 sensors-22-02542-t003:** Comparison of mmWave radar, lidar, and camera [36,37,38,39,40]. “1”–“6” denote the levels from “extremely low” to “extremely high”.

Sensor Type	mmWave Radar	Lidar	Camera
Range resolution	4	5	2
Angle resolution	4	5	6
Speed detection	5	4	3
Detection accuracy	2	5	6
Anti-interference performance	5	5	6
Requirements for weather conditions	1	4	4
Operating hours	All weather	All weather	Depends on light conditions
Cost and processing overhead	2	4	3

**Table 4 sensors-22-02542-t004:** Summary of the three fusion levels.

Fusion Level	Advantages	Disadvantages
Data level	Minimum data loss and the highest reliability	Dependence on the number of radar points
Decision level	Making full use of sensor information	Modeling the joint probability density function of sensors is difficult
Feature level	Making full use of feature information and achieving best detection performance	Complicated computation and overhead of radar information transformation

**Table 5 sensors-22-02542-t005:** Summary of data level fusion.

		Reference	Contribution
ROI generation		[42]	Using radar points to increase the speed of ROI generation.
		[45]	Proposing the conclusion that distance determines the initial size of ROI.
		[54]	Extending ROI application to overtaking detection.
Object detection	Image preprocessing	[45,56,61]	Using histogram equalization, grayscale variance and contrast normalization to preprocess the image.
		[53,57,61]	Image segmentation preprocessing with radar point as reference center.
	Feature extraction	[55,57,58,59,61,63]	Using features such as symmetry and shadow to extract vehicle contours.
		[56,64]	Using Haar-like model for feature extraction.
	Object classification	[56]	Adaboost algorithm for object classification.
		[47,60]	SVM for object classification.
		[61,62]	Neural network-based classifier for object classification.

**Table 6 sensors-22-02542-t006:** Summary of decision level fusion.

		Reference	Contribution
Sensing information processing	Radar information	[67,68]	The techniques involved in radar signal processing and what physical states can be obtained from radar information are analyzed.
	Image object detection	[68]	Pedestrian detection using feature extraction combined with classifiers.
		[69]	Detecting objects in depth images with MeanShift algorithm.
		[70]	An upgraded version of [69], using MaskRCNN for target detection.
		[71,72,80]	Using one-stage object detection algorithm YOLO for radar vision fusion object detection tasks.
Decision fusion	Based on Bayesian theory	[73]	Proposing Bayesian programming to solve multi-sensor data fusion problems through probabilistic reasoning
		[74]	A dynamic fusion method based on Bayesian network is proposed to facilitate the addition of new sensors.
	Based on Kalman filter	[75]	Proposing a decision level fusion filter based on EKF framework.
		[76]	The proposed fusion methon can track the object simultaneously in 3D space and 2D image plane.
		[77]	Functional equivalence of centralized and decentralized information fusion schemes is demonstrated.
	Based on Dempster Shafer theory	[68]	A decision level sensor fusion method based on Dempster-Shafer is proposed.
	Based on Radar validation	[78]	Using radar detection results to validate visuals.
		[79]	Using radar information to correct vehicle position information in real time to achieve object tracking.

**Table 7 sensors-22-02542-t007:** Summary of feature level fusion.

	Reference	Technology Features
Fusion framework	[81]	Based on SSD framework improvement, concatenation fusion is used.
	[82]	A fusion framework similar to YOLO structure is proposed named RVNet.
	[83]	Proposing CRF-Net built on the VGG backbone network and RetinaNet, and the radar input branch is extended.
	[84]	Join the radar branch based on the FCOS detection framework and embedded SAF module.
Radar feature extraction	[85]	Proposing a network named CMGGAN that can generate environmental images.
	[82,84]	Using a new radar feature description method called radar sparse image, the detected objects are presented as radar points.
	[83]	Stretching the radar points in the radar sparse image vertically to supplement the height information.
Feature fusion	[81,82,83]	The fusion method of concatenation and element-wise addition is adopted.
	[84]	A feature fusion block named spatial attention fusion is proposed that uses attention mechanism.

## Data Availability

Not applicable.

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
