# Peer review of "MmWave Radar and Vision Fusion for Object Detection in Autonomous Driving: A Review"

_sensors, 2022, doi:10.3390/s22072542_

Round 1
Reviewer 1 Report
This paper presents an overview of object detection in autonomous driving with a special focus on the fusion of MmWave Radar and Vision. I have read this paper with great interest. However, I am concerned about some issues.
Therefore, the following problems in the current version need to be solved by authors.
(1). This paper lacks a deeper analysis of existing methods as an overview paper. The fusion methods are classified into data level, decision level and feature level fusion methods and some works are listed. However, it would be better to also present what are the challenges from the fusion perspective, and then present the works that are designed to address the corresponding challenges.
(2). This paper needs a better description regarding the big picture, and the relationship regrading big picture and the specific problem. Why autonomous driving is still an open problem? And what role is object detection playing in the big picture of autonomous driving? Will mmWave radar and vision be the ultimate solution to object detection in autonomous driving?
(3). Please provide better quality and comprehensive captions regarding every figure. For example, the Figure 3 is of low quality and what is the meaning of dots with different colors?
(4). Besides including the classical research, please focus more on latest research. A lot of papers are before year 2010, which can not provide up-to-date progress. For example, in line 312-313, I do not think references [50][55] from year 2008 and 1999 can be called "latest literature".
(5). Figure 4 - 11 paste the network structure from different papers. However, simply copy and paste the images doesn't provide any information. Please include comprehensive caption, and explain why you need these figures in this manuscript. Otherwise, please delete these images.
(6). Please double check the copyright problem regarding using the figures directly cropped from other people's paper regarding Figure 4 - 11.
(7). Please include open problems in the fusion of mmWave radar and vision, and future promising directions to address them.
(8). Please go through the paper carefully and fix the typos. For example, in line 1, the initial letter should be capitalized.
Reviewer 2 Report
Thank you for delivering a nice manuscript related to mmWave radar and vision fusion for vehicle detection. The abstract and conclusion are precise and comprehensive discussion has been provided in provided sections. However, there are several issues which authors need to revise as follows.
- Authors should clearly define the abbreviations at the first place of appearance such as HOG, DPM, YOLO, SSD, R-CNN, GPS, V2X, FFT etc. or you can add a nomenclature section.
- Introduction section is nicely written and sufficient references are provided.
- Line 127-129, proper reference must be cited for this provided information.
- Line 179-182, authors have mentioned the advantages of the fusion of mmWave radar and camera, how about fusion with LiDAR? Any specific drawbacks under severe environmental conditions?
- It is suggested to remove * which denotes the levels in Table 3 and add clear words to make it easier for reader.
- Section 4.2 and 4.3.Authors needs to add more significant discussions.
- Table 5. Insufficient details provided under contribution by each listed study.
- At several places, authors have listed references without providing any sufficient discussions.
- Section 5.2 and 5.3, authors have provided detailed explanation about Decision Level Fusion and Feature Level Fusion from several reported works. It will be more interesting if make a summarize or comparative table so the reader can fastly know these contributions.
- It is suggested that a separate section must be dedicated to open research challenges.
- I have noticed a clear lack in details associated to diagrams. Most of the diagrams have been added without providing sufficient information and explanation.
- I found some references are not taken from primary resources such as journals, proceedings, books etc. In my opinion, sources from personal website, blogs are not considered as academic references. Except there are statements regarding to some of these resources.
Round 2
Reviewer 1 Report
Thanks for the response. I appreciate the authors have taken some of my suggestions and the manuscript has indeed improved.
Reviewer 2 Report
Thank you for your explanation and relevant corrections in revised manuscript. You have addressed all my concerns very well in this draft. The quality of presentation, as well as the scientific depth of the paper have been substantially improved. I have no further questions regarding the publication of this manuscript.